# Genome-wide association mapping identifies yellow rust resistance loci in Ethiopian durum wheat germplasm

Sisay Kidane Alemu[1]*, Ayele Badebo Huluka[2], Kassahun Tesfaye[3], Teklehaimanot Haileselassie[4], Cristobal Uauy[5]

1 National Agricultural Biotechnology Research Center, Holeta, Ethiopian Institute of Agricultural Research, Addis Ababa, Ethiopia, 2 International Maize and Wheat Improvement Center (CIMMYT), Addis Ababa, Ethiopia, 3 Institute of Biotechnology and DMCMB, Addis Ababa University, Addis Ababa, Ethiopia, 4 Institute of Biotechnology, Addis Ababa University, Addis Ababa, Ethiopia, 5 John Innes Centre, Norwich Research Park, Norwich, United Kingdom

* sisukidan@gmail.com

**Data Availability Statement:** All relevant data are within the manuscript and its Supporting Information files.

## Abstract

Durum wheat is an important cereal grown in Ethiopia, a country which is also its center for genetic diversity. Yellow (stripe) rust caused by *Puccinia striiformis* fsp *tritici* is one of the most devastating diseases threatening Ethiopian wheat production. To identify sources of genetic resistance and combat this pathogen, we conducted a genome wide association study of yellow rust resistance on 300 durum wheat accessions comprising 261 landraces and 39 cultivars. The accessions were evaluated for their field resistance using a modified Cobb scale at Meraro, Kulumsa and Chefe Donsa in the 2015 and 2016 main growing seasons. Analysis of the 35K Axiom Array genotyping data of the panel resulted in a total of 8,797 polymorphic SNPs of which 7,093 were used in subsequent analyses. Population structure analysis suggested two groups in which the cultivars clearly stood out separately from the landraces. Eleven SNPs significantly associated with yellow rust resistance were identified on four chromosomes (1A, 1B, 2B, and 5A) which defined at least five genomic loci. Six of the SNPs were consistently identified on chromosome 1B singly at each and combined overall environments which explained 62.6–64.0% of the phenotypic variation ($R^2$). Resistant allele frequency ranged from 14.0–71.0%; Zooming in to the identified resistance loci revealed the presence of disease resistance related genes involved in the plant defense system such as the ABC transporter gene family, disease resistance protein RPM1 (NBS-LRR class), Receptor kinases and Protein kinases. This study has provided SNPs for tracking the loci associated with yellow rust resistance and a diversity panel which can be used for association study of other agriculturally important traits in durum wheat.

## Introduction

Durum wheat (*Triticum turgudum* subsp. *durum*) is a tetraploid wheat (2n = 4x = 28) with AABB genome designation and basic chromosome number x = 7. Tetraploid wheat is thought

**Funding:** CU received funding from the UK Biotechnology and Biological Sciences Research Council (BBSRC's) Sustainable Crop Production Research for International Development (SCPRID) initiative (Grant Ref: BB/J012017/1, https://bbsrc. ukri.org/news/topic/scprid). SKA received financial support for the PhD study to conduct 2 seasons field research in Ethiopia from Ethiopian Institute of Agricultural Research (EIAR: http://www.eiar.gov. et/index.php/en) and Addis Ababa University (AAU: http://www.aau.edu.et/) and a 1 year of lab work staying at the JIC, Norwich, UK through the Grant. The funders had no role in study design, data collection and analysis, decision to publish, or preparation of the manuscript.

**Competing interests:** The authors have declared that no competing interests exist.

to be the result of a genome hybridization between the diploid A genome wheat species *Triticum urartu* (2n = 2x = 14) and an S genome species related to *Aegilops speltoides* [1].

Durum wheat is an important food crop with global production estimated to be 36 million t per year [2]. Ethiopia is considered as a center of secondary diversity for durum wheat and as such has a wide array of untapped tetraploid germplasm [3–9]. Although statistics on current production are combined (in most cases) into a single "wheat" category, occasional studies report that Ethiopia is the largest producer of durum wheat in sub-Saharan Africa with approximately 0.6 million hectares (Evan School Policy Analysis and Research [10]. Particularly, landraces (i.e., a locally adapted line that has not been modified through a breeding programme) are known to cover about 70% of the total durum wheat area [11]. The crop is grown widely on heavy black soils of the highlands with an altitude range of 1800 to 2800 m asl [12, 13]. Due to this continuous production by farmers in the highlands, Ethiopian durum wheat has evolved in its phenotypic [14] and molecular diversity [15]. Selection pressure from the natural and artificial sources, inevitably has contributed to the development of many adaptive traits including disease resistance [16–18] which provide an opportunity for genetic improvement programs.

Durum wheat is mostly grown as a food crop by small holder farmers, although it has potential as an industrial crop in the food industries. Generally, pasta, spaghetti, biscuits, pancakes, macaroni, pastries, and unleavened breads are among the known forms of uses of durum wheat [19]; and global durum wheat use is trending upward [20]. Traditionally, 'kinche' (crushed kernels, cooked with milk or water and mixed with spiced butter) is among the most commonly used Ethiopian recipe [21]. Apart from that, in the mixed farming systems of the highlands, the straw of durum wheat is also of high relevance for animal feed due to its digestibility and palatability characters [22].

Stripe/yellow rust of wheat is an obligate biotrophic fungal disease caused b*y Puccinia striiformis* f. sp *tritici* (*Pst*) [23], and one of the most devastating diseases in wheat growing regions of Ethiopia [24]. In 2010, an epidemic of the disease occurred in the country and reached all the wheat growing regions at unprecedented rates. It infected large production areas including 29 zones in the Amhara, Oromia, and Southern Nations, Nationalities, and Peoples (SNNP) regional states [25, 26]; http://www.addisfortune.com.

Stripe rust can be managed using various methods such as cultural practices, fungicide application, and resistance cultivar development and deployment [27–30], with the latter being an environmentally friendly strategy. With regard to exploiting genetic resistance, worldwide research reports on wheat rusts so far indicate that more than 80 *Yr* genes (*Yr1-Yr83*) have been permanently named which are known to confer both adult plant resistance (APR) and/or all stage (seedling resistance) [31–34]. Most of these genes originally derived from hexaploid wheat (*Triticum aestivum* and their wild relatives); some of them from diploid wild relatives and very few of them (such as *Yr15* and *Yr36*) from tetraploid wild relatives (*Triticum dicoccoides*) wihle just *Yr24* from durum wheat (*T. turgidum*). A partial List of catalogued *Yr* genes derived from various sources is summarized in S1 Table. Resistance genes having diagnostic markers can be transferred to well adapted but susceptible cultivars through Marker Assisted Selection (MAS) coupled with gene pyramiding strategy. For instance, *Yr* genes such as *Yr26*, *Yr51*, *Yr57*, *Sr* genes like *Sr22*, *Sr26* and *Sr50* have been transferred to well adapted cultivars through Marker Assisted backcrossing method which is one of MAS approaches [35–37]. Due to its dynamic nature, new virulent *Pst* races often emerge and break varietal resistance by defeating the genes they carry and drive them out of production. This situation prompts the necessity of continuous search and identification of new sources of resistance to sustain wheat production.

In Ethiopia, research effort made by the wheat improvement programs has resulted in the development of cultivars with relatively good levels of wheat rust resistance, good yield, and desirable agronomic characters [38]. Germplasm screening to identify resistant breeding lines and development of elite durum wheat cultivars [39] has been part of the strategies to exploit the host's natural defence in breeding programs. Screening available germplasms for the presence of those known *Yr* genes is one of the strategies to maximize the resistance potential of that germplasm. Specifically, when the genes are mapped to a known genomic locus and diagnostic molecular markers are available [40], the detection of the known resistant genes is facilitated. However, several of the identified genes have limited spectra of effectiveness as they have been previously overcome by specific *Pst* races. Besides, germplasm pools vary in their genetic background and local adaptation in which case deciphering the resistance genes they carry is difficult specifically in the absence of perfect markers. One strategy to address this is searching for resistance genes in locally adapted germplasm using techniques such as Genome Wide Association Studies (GWAS). GWAS studies ultimately result in the identification of genomic loci/SNPs which are significantly associated with the target traits [41] and can be used in marker assisted introgression programs to improve the target crop. As a follow-up investigation, the associated SNPs can be validated for their diagnostic values in an independent germplasm.

Ethiopian durum landraces have been suggested as a good source of resistance to wheat rusts [42, 43]. This has been supported by association studies on relatively small sized Ethiopian durum panels in limited study environments [17]. Extensive research on potential new and effective *Pst* resistance associated loci or genes is necessary to cope with the occurrence of new virulent *Pst* races. The availability of high-density SNP wheat chips e.g., 35k Breeders chips, iSelect 9 and 90 K SNP assays [44–46] provides an opportunity to facilitate such genome level studies. Therefore, the objectives of this study were: (1) to access the extent of field resistance to stripe rust among Ethiopian durum wheat panel and (2) to identify SNP markers/ genomic loci significantly associated with Stripe rust resistance.

## Materials and methods

### Phenotyping

**Plant materials.**   Durum wheat accessions (n = 513) were obtained from various sources in Ethiopia. The accessions comprise landraces from gene bank collections (Ethiopian Biodiversity Institute (EBI) and Ethio-Organic Seed Action (EOSA)) and few additional landraces and cultivars (released varieties) from Debre Zeit Agricultural Research Center (DZARC). To start with a relatively pure seed stock, all the accessions were grown and those having a mixed seed stock were identified by physical observation on morphological features such as spike architecture (e.g., density, color, awns) and seed color. In few instances, accessions of mixed genotypes were split and considered as a separate accession making the number a bit increased. Each accession was then subjected to a single spike row planting followed by two generations of self-pollination. A final single spike to row planting was carried out at Holeta Agricultural Research Center for seed multiplication. The final working population was then constituted and cut down to 300 based on morphological similarity among the accessions and used for phenotypic evaluations and genotyping. This final panel included 261 landraces and 39 cultivars. The source and related description of the accessions is provided in S2 Table.

**Field resistance evaluation.**   The accessions were grown in an alpha lattice design with two replications at Chefe Donsa (CHD), Kulumsa (KUL) and Meraro (MER) testing sites in the main growing season (June–November) of 2015 and 2016. Chefe Donsa is located 35 kms east of Debre Zeit at 08˚57'15" N and 39˚06'04" E and has an altitude of 2450 m. Kulumsa is

located 167 kms from Addis Ababa at 8°01'11.7"N 39°09'38.2"E and has an attitude of 2200 m; whereas Meraro is located about 236 kms from Addis Ababa at 7°24'25.8"N 39°14'56.3"E and has an elevation about 3,030 m. Each accession in each replication was sown in two rows of 0.5 m length, with 0.2 m spacing between rows. Each block was enclosed between spreader rows of known susceptible durum wheat (LD-357, Arendato and Local Red) and bread wheat (Morocco known to have *Sr25* and *Lr19*) varieties mixed in the ratio of 1:1:1:1 and sown 20 cm from the experimental plots. The spreader rows act as an inoculum source and help in achieving uniform disease establishment throughout the experimental field. The disease severity (percentage of leaf tissue infected with the rust) was evaluated using a modified Cobb's scale [47] with values ranging from 0 to 100%. The field response of the genotypes to the rust infection was scored according to Stubbs et al. [48] as R (resistant), MR (Moderately Resistant), Moderate (M), MS (Moderately, Susceptible) and S (Susceptible) each having a numerical constant value of 0.2, 0.4, 0.6, 0.8 and 1.0, respectively. The scoring or field resistance was usually commenced when disease severity is greater or equal to 50% and a susceptible reaction (S) was observed on the spreader rows which is presented as a combined value 50S. All agronomic practices were applied following the recommended practice for wheat at each location. Two data sets (one for each year) per each of the three locations was generated giving a total of six environments.

**Phenotype data analysis.** Severity (SEV) score was multiplied with the field Response (RES) values to produce the Coefficient of Infection (CI) which represents the combined reaction of the genotypes for the pathogen. The panel was classified in to Resistant, Intermediate and Susceptible groups based on the average values of SEV and RES [17, 49] for each environment. As CI is the product of SEV and RES, the corresponding values were used to do the same reaction classification in terms of CI as well. To comply with the normality assumption, SEV and CI data were transformed with common logarithmic function ($SEV_{tr} = LOG_{10}(SEV +1)$ and $CI_{tr} = LOG_{10}(CI+1)$) while Response was subjected to arcsine transformation ($RES_{tr} = arcsine(\sqrt{RES})$). The level of infection on some of the known susceptible genotypes was assessed for its consistency of susceptibility and used as a basis to judge the suitability of the disease reaction data across environments for downstream analysis. Accordingly, level of infection data of CHD-2015 and KUL-2015 were found to be so low to be reliable and excluded from all downsteam analyses. Shapiro Wilk test (Shapiro and Wilk 1965 [50]) was applied on the original and the transformed data to assess the normality of the data. Analysis of Variance (ANOVA) was done with the transformed data for each and combined over environment using a linear mixed model. In the model, genotype, Incomplete block within Replication, Location, year, and genotype by other variance source interactions were considered as random effects. Multiple random effect test (Variance component test) was carried out using Lym4 and LimerTest packages in R. The Best Linear Unbiased Estimates (BLUEs) of each environment and combined data (BLUE-all) was generated using Restricted Maximum Likelihood (REML) method fitting the genotype as fixed effect while the other variance sources and interactions as random effect. These BLUE values were used to perform the association analysis. Correlation analysis was performed for SEV, RES and CI to assess the extent of covariation of the resistance reaction within and across environments.

## Genotyping

Seeds were sown on a 50 mm diameter sterile petri dish equipped with 42 mm filter discs to maintain a moist condition for germination. After watering them with distilled water, petri dishes were incubated at 4°C under dark conditions for 24 hours to break dormancy. The dishes were then kept at room temperature for 3–4 days until fully germinated. Germinated

seeds were planted in a 96 well tray filled with peat and sand soil mix optimized for raising cereal seedlings. The trays were placed in cereal growth chamber set at 19˚C day and 16˚C night temperature with a relative humidity of 70% and a photoperiod of 16/8 hours light/dark cycle.

Fully opened leaves were harvested from 10 to 14 days old seedlings in a 1.2 ml deep-well plate on a dry ice and freeze-dried for 24–30 hours at -40˚C under a pressure of 20 atm. The tissue was then ground into fine powder followed by a wet grinding with Geno/Grinder 2010 at 1750 rpm for 2 minutes. Genomic DNA was extracted with SDS buffer following the wheat and barley DNA extraction protocol in 96-well Plates [51] with some modification. DNA was cleaned according to Affymetrix User Guide, Axiom® 2.0 Assay for 384 Samples (Genomic DNA Preparation and Requirements) and quantified with Nanodrop (8-sample spectrophotometer ND-8000). A total of 100 µL of DNA sample normalized to 75–100 ng/µL was submitted to University of Bristol Genomic Facility for genotyping using Breeders' 35K Axiom Array. At the service center, sample array processing and genotyping was carried out following the Axiom® 2.0 Assay for 384 Samples user and Workflow guide (http://media.affymetrix.com /support/downloads/manuals/axiom_2_assay_auto_workflow_user_guide.pdf). We received the genotype data as ARR, JPEG and CEL files, where the later was used for SNP and all other downstream analyses.

## Analysis

**Genotype/SNP data analysis.** Genotype/SNP analysis was performed with Axiom Analysis Suit (AxAS) v2.0 using the CEL intensity files following sample QC and the Best Practice Workflow. Poor quality samples were identified with Dish Quality Control (DQC) values where samples having a value of $\leq 0.80$ were excluded from the next step of the analysis. Using a subset of probe sets, samples which passed the DQC value step were subjected to genotype calling to generate the QC call rate and those samples which had a value $\leq 0.91\%$ were excluded from all subsequent genotyping and SNP data analysis. Once the good quality (QC passed) samples were identified, the downstream genotype data analysis was performed following the Best Practice Workflow with SNP QC default settings for SNP genotype calling. The resulting SNP classes were assessed for complying with expected thresholds mainly with the number of minor alleles $\geq 2$. Accordingly, the 'polyhighresolution' and 'NoMinorHomos' classes, as they fulfill the criteria, were closely examined, and considered for the next step analysis. The other SNP classes (MonoHighResolution, OTV, Other and CallRateBelowThreshold) were not considered because they are noisy in many aspects and did not comply with the minor allele number threshold mentioned above.

The SNP summary table and the genotype call data were extracted using the export tab of the Axiom analysis window. The accessions' genotype data was further subjected to individual heterozygosity analysis and accessions with a value of $\geq 3\%$ heterozygosity were excluded from the panel. The physical positions of the SNPs were extracted from the position file of Breeders' 35K Axiom Array anchored on the wheat reference genome sequence RefSeq v1.0 [52]. Minor Allele Frequency (MAF) was calculated as a percentage of each SNP allele relative to the total in the association panel and individuals having a MAF value > 5% (cut-off). Furthermore, Locus Heterozygosity (the probability that an individual is heterozygous for the locus in the population and polymorphism Information content (PIC: the discriminating power of the marker in a population) were calculated as described in Liu [53] and Guo [54]. With all the threshold values of heterozygosity and MAF, the final working number of the panel was determined 293 while that of the working SNP data was cut down from 8797 to 7093 and these were used in all downstream analyses.

**Population structure & kinship.** Population structure among the association panel was analyzed using STRUCTURE v 2.3.4 [55]. Allele frequency model was employed at Length of burnin period = 10,000, number of MCMC Reps after burnin = 100,000, K runs from 1–10 with 5 replications for each K. The result of the analysis was zipped into a folder and uploaded on to STRUCTURE HARVESTER [56], an online analysis tool used to generate an estimate of the optimum number of subpopulations (i.e., the value of K) through Evanno method [57]. Kinship matrix was generated through genetic similarity matching using all possible pairwise combination among the panel using the R package Genomic Association and Prediction Integrated Tool (GAPIT) besides performing the PCA [58].

**Linkage disequilibrium.** Genome-wide Linkage Disequilibrium was assessed using TASSEL v.5 to estimate the squared allele frequency correlation ($r^2$) for all pairwise comparison of distances between SNPs. The genotype data was imported into TASSEL in HAPMAP format and full matrix LD analysis was performed with the default settings. The resulting LD output was subjected to binning using customized R script with bin_size of 1,000 bp and maximum_bin of 829,100,000 bp to generate the average $r^2$ and the inter-SNP distance values. The LD values were plotted against the corresponding inter-SNP physical distance using a customized R script to estimate the LD decay rate as described in Hill and Weir [59]. The threshold value of the LD ($r^2$) was set at 0.2 a commonly applied value in many related studies [60–63]. To visualize the LD decay pattern, nonlinear model was fitted to the LD plot relating the squared allele frequency with the physical distance [64].

**Association analysis and locus exploration.** The working set 7093 SNP genotype data across the 293 genotypes was organized in the HapMap format. The phenotype data in terms of the Best Linier Estimate (BLUE) for SEV, RES, and CI was arranged across the 293 genotypes for all the single environment data sets. GAPIT [58] was used for the association analysis in R v 4.0.3 and RStudio v. 1.4.1103 to identify the genomic loci underlying the *Pst* resistance. The markers and the phenotype data corresponding to each genotype was input and fitted in the compressed mixed linear model (MLM) which accounts for uneven relatedness and controls it effectively through lowering the type I errors [65]. VanRaden's method [66] was applied to calculate Kinship. To account for the genetic structure, Principal Component Analysis (PCA) was calculated and iteratively added to the model [67]. The best fit of the model was visually assessed by observing the QQ plots. Correction for false positive association was then performed by adding the default K + PCA covariates to the fixed effect part of the model in the GAPIT code. The analysis was carried out first at each environment using a single dataset, then combined over all environments. The probability of adjusted false discovery rate ($p < 0.05$) was used as a critical value to declare the significant marker trait association [68]. The "-log10(p)" values were plotted against each chromosome and the "expected -log10(p)" to generate Manhattan and QQ-plots respectively with codes imbedded in the GAPIT script. The SNP allele in the most and consistently susceptible line EDW-262 was used as the susceptible allele and the alternative was considered as the resistant one. The resistant allele frequency among the panel was determined based on the number of individuals which harbor the resistant alleles at each locus.

Single locus-based variation of phenotypic values of resistant and susceptible alleles was also tested using Two-Sample T-test assuming equal variances. Further exploration was done on genomic regions on chromosomes encompassing all identified loci significantly associated with the resistance trait. Genes and their functional descriptions reported with in the identified resistance associated regions and close to the nearest significantly associated SNP were examined using list of genes/gene model extracted from the wheat genome annotation file [52].

## Results

### Phenotypic reaction to Pst infection and distribution

Variably distributed reaction groups were demonstrated among the panel for SEV, RES and CI across the six environments (Fig 1A–1C). For SEV, 100% of the accessions were classified as resistant (0≤SEV≤10) at CHD_15 and 99.7% at KUL_15 while 49, 95, 84, 58% were classified at MER_15, CHD_16, KUL_16 and MER_16 respectively (Fig 1A). For RES, 96.6% of the accessions fell under resistant class at CHD_15 and 97.63% at KUL_15, while 33, 91, 79, 41% were observed under the same resistant class at MER_15, CHD_16, KUL_16 and MER_16 in

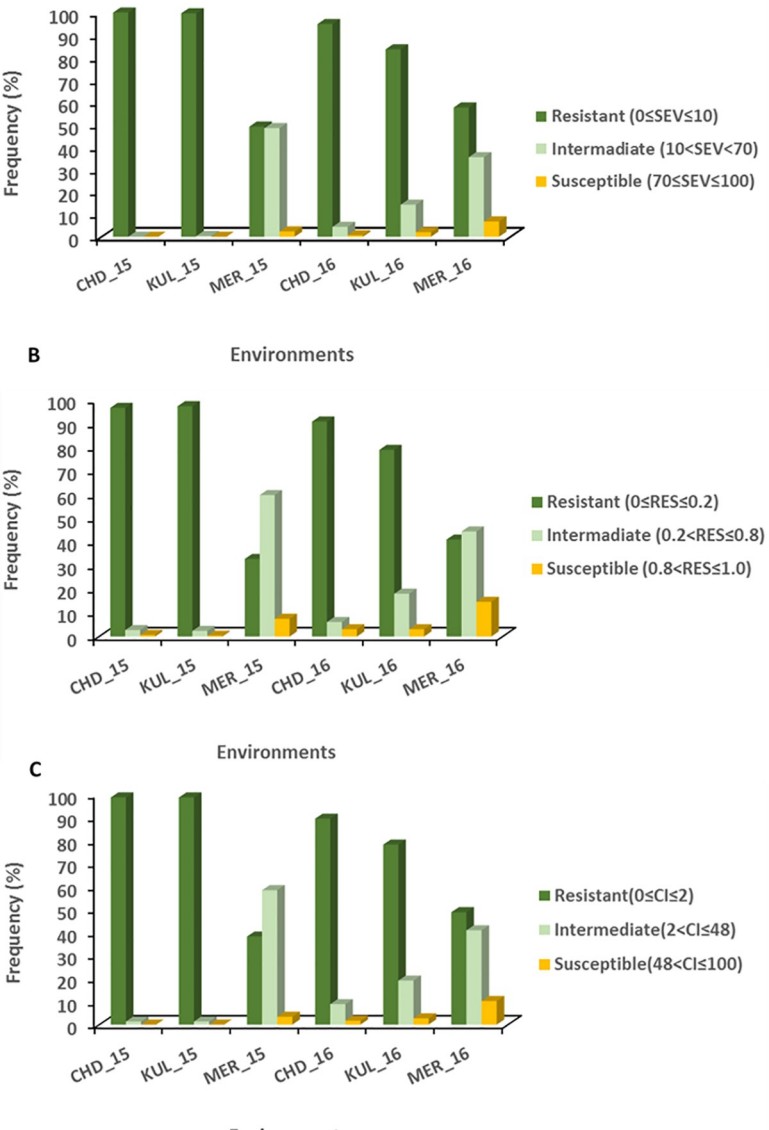

**Fig 1. Reaction to yellow rust of 293 Ethiopian durum wheat accessions obtained from field experiments in six environments.** A) severity, B) Field Response and C) Coefficient of Infection. The reaction data of severity and field response was used to classify the panel in to Resistant, Intermediate and Susceptible groups as described in Liu et al. [17] and taking the corresponding values for Coefficient of Infection as well.

that order (Fig 1B). Almost similar frequencies of reaction groups were observed for CI where 99% of them were under resistant class both at CHD_15 and KUL_15 while 38, 89, 78, 49% appeared in the same reaction group respectively at MER_15, CHD_16, KUL_16 and MER_16 (Fig 1C). The reaction data of each genotypes of the panel to *Pst* has been presented in S2 Table. For all the SEV, RES and CI values of CHD-15 & KUL-15, almost 100% of the panel shown overall resistant reaction unlike their reaction at CHD-2016, KUL-2016, MER-2015 and MER-2016. Besides, investigation of the reaction of the five selected susceptible genotypes (Table 1) across the six environments showed that an intermediate or resistant reaction for CHD-15 and KUL-15. This is too low infection level to be considered in further analysis as it affects results and leads to erroneous conclusion. So based on this assessment, the data of those two environments were excluded from all downstream analyses. The mean values of SEV, RES and CI combined over the remaining four environments was 10.6%, 0.3 and 8.1 in the same order. Considering stability of reactions across all the four environments, 37.2, 25.9, and 31.1% of the genotypes gave consistently resistant reaction for SEV, RES and CI across the testing location in that order (S3 Table). Because CI is a combined representation of SEV and RES for the resistance reaction, 31.1% (91 genotypes) of the total 293 panel were considered as stably resistant across the four environments. The list of these resistant genotypes along with the SEV, RES and CI values can be accessed from S2 Table.

Distribution of the untransformed and transformed data is presented in Fig 2A–2F. Despite discarding CHD-15 and KUL-15 data from the analyses, the frequency distribution of the original untransformed data (BLUE-all-Original) values was still skewed for SEV (W = 0.67) and CI (W = 0.58), while RES appeared to be less skewed (W = 0.91) (Fig 2A–2C). Application of logarithmic transformation to SEV and CI and the arcsine transformation to RES changed the distribution to better adjusted normality with W values of 0.93 for SEV, 0.93 for RES and 0.88 for CI (Fig 2D and 2F).

## Variances and correlations of reactions

We first examined the variance components for the multiple factors included in the statistical model. ANOVA of each environment and combined across environments is summarized in Table 2. Genotypic variance component was significant (*P < 0.001*) for all the individual environments and combined across environments (Table 2). Except for MER-15 & MER-16, Blocks nested within replications were non-significant for CHD, KUL and combined across environments. Similarly, except for year variance components of locations and the two-way genotype by environment interactions (GxY & GXL) were significant at *P < 0.001* for all SEV, RES & CI. Likewise, the three-way interaction (GxLxY) resulted in a highly significant variation at *P < 0.001* for SEV, RES and CI. Apparently, there was strong interaction between the response of each genotype based on the location and year, which is consistent with the result presented in Fig 1A–1C).

Next, we calculated the correlation coefficients to see the relationships between the three phenotypic variables within and across environments (Table 3). Within MER, correlation coefficient (*r*) between data of 2015 & 2016 was 0.70 for SEV; 0.72 for RES and 0.68 for CI. whereas the data from CHD and KUL was only for 2016, and no correlation was calculated for within location level. On the other hand, overall correlation among all four environments was 0.67 ± 0.05 for SEV vs SEV, 0.66 ± 0.05 for RES vs RES and 0.67± 0.05 for CI vs CI (Table 3). Correlations among the three reaction data types was also assessed at different combinations (SEV vs RES, SEV vs CI & RES vs CI). High correlation was observed for SEV vs RES (*r* = 0.94 ± 0.01), SEV vs CI (*r* = 0.97 ± 0.01) and RES vs CI (*r* = 0.93 ± 0.01) within same environment while very low and nearly similar for within same location- different year and among

**Table 1. Severity, field response and coefficient of infection for susceptible genotypes as measure of infection level across the six environments.**

| Genotypes[a] | Severity | | | | | | Field Response | | | | | | Coefficient of Infection | | | | | |
|---|---|---|---|---|---|---|---|---|---|---|---|---|---|---|---|---|---|---|
| | CHD15 | CHD16 | KUL15 | KUL16 | MER15 | MER16 | CHD15 | CHD16 | KUL15 | KUL16 | MER15 | MER16 | CHD15 | CHD16 | KUL15 | KUL16 | MER15 | MER16 |
| EDW_299 | 3.0 | 80.0 | 0.0 | 100.0 | 100.0 | 100.0 | 0.9 | 1.0 | 0.0 | 1.0 | 1.0 | 1.0 | 2.9 | 80.0 | 0.0 | 100.0 | 100.0 | 100.0 |
| EDW_276 | 0.0 | 50.0 | 0.0 | 85.0 | 85.0 | 95.0 | 0.0 | 0.9 | 0.0 | 0.9 | 1.0 | 1.0 | 0.0 | 46.0 | 0.0 | 78.0 | 85.0 | 95.0 |
| EDW_262 | 0.0 | 60.0 | 12.5 | 100.0 | 85.0 | 100.0 | 0.0 | 1.0 | 1.0 | 1.0 | 1.0 | 1.0 | 0.0 | 60.0 | 12.5 | 100.0 | 85.0 | 100.0 |
| EDW_263 | 2.5 | 65.0 | 0.5 | 70.0 | 75.0 | 95.0 | 0.5 | 0.9 | 0.1 | 1.0 | 0.9 | 1.0 | 2.5 | 60.0 | 0.1 | 70.0 | 70.0 | 95.0 |
| EDW_264 | 0.0 | 75.0 | 0.0 | 90.0 | 97.5 | 100.0 | 0.0 | 1.0 | 0.0 | 1.0 | 1.0 | 1.0 | 0.0 | 75.0 | 0.0 | 90.0 | 97.5 | 100.0 |
| Average | 1.1 | 66.0 | 2.6 | 89.0 | 88.5 | 98.0 | 0.3 | 1.0 | 0.2 | 1.0 | 1.0 | 1.0 | 1.1 | 64.2 | 2.5 | 87.6 | 87.5 | 98.0 |

[a] EDW: Ethiopian Durum Wheat; of the five genotypes, EDW_299 (a known susceptible cultivar used in infection experiments named as Local Red) is also included in the spreader row mix.

CHD: Chefe Donsa; KUL: Kulumsa; CI: Coefficient of Infection.

For all disease score values of Severity, Filed Response and Coefficient of Infection: values highlighted in Gray represent **Resistant** reaction; values in bold font represent **Intermediate** reactions; and values in normal font represent **Susceptible** reaction.

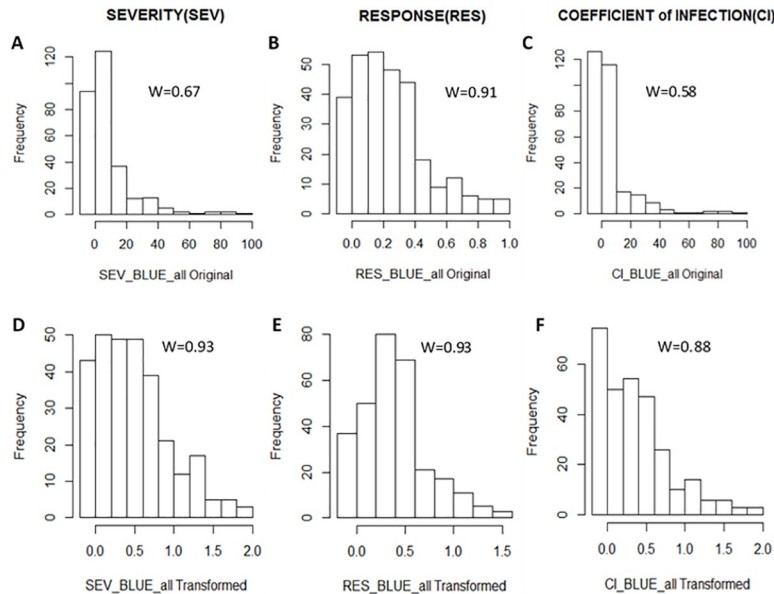

**Fig 2. Distributions of disease severity (SEV), response (RES) and coefficient of infection (CI) combined over four environments.** A, B and C represents distributions of best linear unbiased estimates (BLUEs) of original for SEV, RES and CI data while D, E and F represents transformed data in their respective order. "W" is Shapiro—Wilks Statistic indicating the correlation between observed values and normal scores both for the original and transformed values.

locations (Table 3). This is expected because the level of disease pressure that result in a high severity is highly likely to result in higher reaction of the genotype exposed and hence positively impact the coefficient of correlation. Besides, using the T-distribution test for most of the comparisons shows, statistically significant correlations at $P < 5\%$ except very few marked as "ns" (Table 3). This can be partly explained by the higher degrees of freedom in the analysis (i.e., DF = 293–2 = 291) and due to the large number of samples.

## Genotypes/SNPs called and chromosomal distribution

Genotype data assessment on the 300 accessions resulted in 296 which passed Dish Quality Control (DQC) $\geq$ 0.8. An example of Cluster plots of the panel resulted from Axiom Analysis Suit V.2 with Best Practice Workflow is presented in Fig 3. Heterozygosity and SNP analysis defined a final panel of 293 accessions with 10,622 polymorphic SNP markers after excluding Individuals with Heterozygosity values of $\geq$ 3%. We extracted the physical positions (from RefSeqv1.0) of the informative SNPs which led to 3853 (36.3%) markers being assigned to the A-genome, 4944 (46.5%) to the B-genome, 481 (4.5%) to the D-genome and 1344 (12.7) to "Chromosome U" (unknown physical position and no significant match in the genome). Excluding the D-genome SNPs (as they are not expected in a tetraploid/durum wheat) and the unknown set resulted in a total of 8797 physically positioned SNPs.

Varying numbers of SNPs, minor allele frequencies (MAF), Locus heterozygosity and polymorphic information content (PIC) value of the SNPs were obtained at chromosome, subgenome, and whole genome levels (Table 4). The highest number of SNPs (852) were observed on chromosome 1B while the lowest (387) was found on chromosome 4A. Mean MAF was the highest (0.2171) for chromosome 1B while it was the lowest (0.1631) for chromosome 2A. Mean locus heterozygosity and PIC values were highest (0.2879 and 0.2326) for chromosome 1B SNPs and lowest (0.2214 and 0.1852) for chromosome 3B respectively. On the average, we identified 628 ± 42 SNPs per chromosome.

**Table 2. Variance[a] in reaction to yellow rust of 293 Ethiopian durum wheat accessions per each environment and combined over environments.**

| Source | Chefe Donsa (CHD-16) | | | Kulumsa (KUL-16) | | | Meraro (MER-15) | | | Meraro (MER-16) | | | Combined Over Environments | | |
|---|---|---|---|---|---|---|---|---|---|---|---|---|---|---|---|
| | SEV | RES | CI | SEV | RES | CI | SEV | RES | CI | SEV | RES | CI | SEV | RES | CI |
| Mean | 2.8 | 0.1 | 2.1 | 7.1 | 0.2 | 4.9 | 15.8 | 0.4 | 11.6 | 16.9 | 0.4 | 13.8 | 10.6 | 0.3 | 8.1 |
| Min | 0.0 | 0.0 | 0.0 | 0.0 | 0.0 | 0.0 | 0.0 | 0.0 | 0.0 | 0.0 | 0.0 | 0.0 | 0.0 | 0.0 | 0.0 |
| Max | 80.0 | 1.0 | 80 | 100.0 | 1.0 | 100.0 | 100.0 | 1.0 | 100.0 | 100.0 | 1.0 | 100.0 | 100.0 | 1.0 | 100 |
| G | 0.1194\*\*\* | 0.0789\*\*\* | 0.0923\*\*\* | 0.2305\*\*\* | 0.1118\*\*\* | 0.2038\*\*\* | 0.2978\*\*\* | 0.1328\*\*\* | 0.2873\*\*\* | 0.3517\*\*\* | 0.1977\*\*\* | 0.3740\*\*\* | 0.1619\*\*\* | 0.0895\*\*\* | 0.1462\*\*\* |
| B(R) | 0.0000 ns | 0.0000 ns | 0.0000 ns | 0.0004 ns | 0.0000 ns | 0.0000 ns | 0.0075\*\* | 0.0024. | 0.0105\*\* | 0.006\*\* | 0.0028\* | 0.0065\*\* | 0.0000 ns | 0.0000 ns | 0.0000 ns |
| L | - | - | - | - | - | - | - | - | - | - | - | - | 0.1090\*\*\* | 0.0690\*\*\* | 0.0767\*\*\* |
| Y | - | - | - | - | - | - | - | - | - | - | - | - | 0.0001 ns | 0.0002 ns | 0.0003 ns |
| G:L | - | - | - | - | - | - | - | - | - | - | - | - | 0.0285\*\*\* | 0.0138\*\*\* | 0.0235\*\*\* |
| G:Y | - | - | - | - | - | - | - | - | - | - | - | - | 0.0364\*\*\* | 0.0107\*\* | 0.0311\*\*\* |
| G:L:Y | - | - | - | - | - | - | - | - | - | - | - | - | 0.0204\*\*\* | 0.0132\*\* | 0.0359\*\*\* |
| ERROR | 0.0516 | 0.0412 | 0.0334 | 0.0941 | 0.0517 | 0.0475 | 0.0871 | 0.0494 | 0.0924 | 0.0591 | 0.0414 | 0.0537 | 0.0741 | 0.0463 | 0.0581 |

[a] Variance represents proportion of the phenotypic values in terms of Severity, Field Response and Coefficient of infection explained by the respective sources of variation; Min = Minimum values; Max = maximum value.

SEV, disease severity; RES, Field Response; CI, Coefficient of Infection (i. e. CI = SEV x RES).

G, genotype variance estimate; B(R), Block nested with in replication variance estimate; L, location variance estimate; Y, year variance estimate; G:L, genotype x location variance estimate; G:Y, genotype x year variance estimate; G:L:Y, genotype x Location x year variance estimate.

ns, not significant. = P < 0.1

\* = P < 0.05

\*\* = P < 0.01

\*\*\* = P < 0.001.

**Table 3. Correlation coefficients of SEV, RES and CI values within and among four environments.**

| SEV vs. SEV | CHD_16 | KUL_16 | MER_15 | MER_16 |
|---|---|---|---|---|
| CHD_16 | 1 | | | |
| KUL_16 | **0.79** | 1 | | |
| MER_15 | 0.53 | 0.66 | 1 | |
| MER_16 | 0.59 | 0.78 | 0.70 | 1 |
| **RES vs. RES** | **CHD_16** | **KUL_16** | **MER_15** | **MER_16** |
| CHD_16 | 1 | | | |
| KUL_16 | 0.76 | 1 | | |
| MER_15 | 0.54 | 0.66 | 1 | |
| MER_16 | 0.56 | **0.77** | 0.72 | 1 |
| **CI vs.CI** | **CHD_16** | **KUL_16** | **MER_15** | **MER_16** |
| CHD_16 | 1 | | | |
| KUL_16 | **0.80** | 1 | | |
| MER_15 | 0.52 | 0.67 | 1 | |
| MER_16 | 0.57 | 0.78 | 0.68 | 1 |
| **SEV vs. RES** | **CHD_16** | **KUL_16** | **MER_15** | **MER_16** |
| CHD_16 | **0.96** | 0.78 | 0.54 | 0.57 |
| KUL_16 | 0.76 | **0.96** | 0.67 | 0.74 |
| MER_15 | 0.53 | 0.64 | **0.91** | 0.68 |
| MER_16 | 0.58 | 0.78 | 0.73 | **0.95** |
| **SEV vs. CI** | **CHD_16** | **KUL_16** | **MER_15** | **MER_16** |
| CHD_16 | **0.96** | 0.84 | 0.58 | 0.62 |
| KUL_16 | 0.72 | **0.96** | 0.71 | 0.80 |
| MER_15 | 0.47 | 0.61 | **0.98** | 0.66 |
| MER_16 | 0.53 | 0.73 | 0.71 | **0.97** |
| **RES vs.CI** | **CHD_16** | **KUL_16** | **MER_15** | **MER_16** |
| CHD_16 | **0.94** | 0.81 | 0.57 | 0.61 |
| KUL_16 | 0.73 | **0.94** | 0.68 | 0.81 |
| MER_15 | 0.49 | 0.62 | **0.91** | 0.69 |
| MER_16 | 0.53 | 0.72 | 0.68 | **0.95** |

Values in grey highlight represent same location but different years comparisons while bold face fonts indicate comparison within same environments.

[ns] all Correlation coefficient values having this superscript represents statistically non-significant correlations between the tested data based on T-distribution.

We further explored the SNPS by looking at the sub-genome distribution. We found lower number of A genome (43.8%) than B genome (56.2%) SNPs (Fig 4). SNPs from both genomes had similar MAF (0.1857, 0.1871), locus heterozygosity (0.2623,0.2634) and PIC (0.2158, 0.2162). Group 1 chromosomes had the highest SNP density/coverage, diverse and informative SNPs. For more reliable result in downstream analyses, these SNPs were further refined to maintain only those with a > 5% MAF cut-off resulting in a final set of 7093 SNPs (Table 4). The distribution of this final SNP set was on average 442.1 ± 27.0 for the A-genome and 571.1 ± 53.0 for the B-genome.

## Population groups and relatedness

We performed a population structure analysis which grouped the panel into two clusters and kinship analysis which revealed further sub groupings (Fig 5A–5F). Landraces clustered

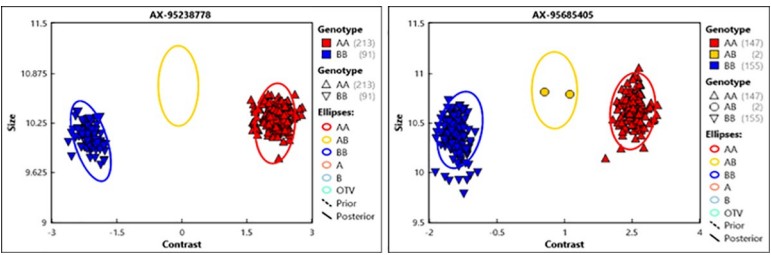

**Fig 3. Examples of cluster plots of durum association panel resulted from axiom analysis suit V.2 with best practice workflow.** QC Threshold passed accessions are 296 and 8 samples are control; AX-95238778 & AX-95685405 are Axiom SNP probes with which the accessions are genotyped.

together into one group while all the cultivars clustered in a second group (Fig 5A). Structure harvester analysis suggested that delta K attained its highest value at K = 2 which is the most likely number of populations in the study panel (Fig 5B). In a global view, kinship analysis resolved the panel into two distinct groups as well, where the cultivars still stood out separately from the rest of the panel (Fig 5C). A closer look at the grouping pattern however revealed the presence of four groups where the cultivars (CL) clustered separately while the landraces separated into three sub-groups (LR-I, LR-II & LR-III). Principal Component Analysis (PCA) also resolved the panel into four clusters with the cultivars noticeably still isolated from the landraces (Fig 5D). Despite PCA revealed four clusters, only the first two components explained 66.67% (Fig 5E) of the variation which is in agreement with presence of two main groups as depicted by structure and kinship analyses.

## Linkage disequilibrium

We computed the squared allele frequency correlation ($r^2$) for all pairwise comparisons of distances between SNPs using TASSEL. At a genome-wide level, 627,886 inter-SNP distances were found through binning LD analysis output at a bin of 1,000 bp. Of this, 176,571 (28.12%) were in significant LD with an average LD estimate of $r^2 = 0.11$ where the highest value (0.28) was achieved within the first 10 Mbp of physical distance (Table 5). The mean LD above the critical value ($r^2 = 0.2$), was also the highest (0.36) within the same 10 Mbp of physical distance as reported for the total (Table 5).

The fitted LOESS curve intersected with the critical LD value at physical distance of 69.1 Mbp where all the values of LD bellow this point were considered to be due to physical linkage among the inter-SNP pairs (Fig 6). The LD started to decay below this critical value to an average $r^2$ of 0.16 for an increase of 10 Mbp (in the interval 10–20) suggesting that the overall LD accounted for the association is exhibited by relatively a shorter genomic distance.

## Marker-trait associations for resistance to yellow rust

The association analysis of CI resulted in a total of 11 SNPs, across four chromosomes (1A, 1B, 2B and 5A) significantly associated with yellow rust resistance at FDR-adjusted *P≤0.05* (Table 6). Two SNPs (AX-94482796, AX-94856684) at CHD and another three SNPs (AX-94438404, AX-94460229, AX-94648330) at KUL were identified as location specific resistance associated SNPs. Six of the 11 identified SNPs (AX-95171339, AX-94436448, AX-95238778, AX-95096041, AX-94730403 & AX-94427201) however, were consistently identified (Table 6 & Fig 7A–7E) at each location and combined analysis over all four environments (BLUE-all). No location specific SNP was identified at MER. Almost similar sets of SNPs were identified from GWAS analysis for SEV and RES as well. As both SEV and RES are highly correlated to

**Table 4. Polymorphism of SNPs in Ethiopian durum wheat germplasm obtained from genotyping with Breeders' 35K Axiom Array.**

| Chr. | No. of SNPs | No Call Rate (Mean) | MAF | | Heterozygosity | | PIC | | No. of SNPs with MAF > 5% [b] |
|---|---|---|---|---|---|---|---|---|---|
| | | | Mean | Range | Mean | Range | Mean | Range | |
| 1A | 548 | 0.0034 | 0.2000 | 0.0034–0.5000 | 0.2766 | 0.0068–0.5000 | 0.2265 | 0.0068–0.3750 | 462 |
| 1B | 852 | 0.0038 | 0.2171 | 0.0017–0.5000 | 0.2879 | 0.0034–0.5000 | 0.2326 | 0.0034–0.3750 | 700 |
| 2A | 615 | 0.0037 | 0.1631 | 0.0017–0.5000 | 0.2399 | 0.0034–0.5000 | 0.2003 | 0.0034–0.3750 | 491 |
| 2B | 882 | 0.0037 | 0.1654 | 0.0017–0.5000 | 0.2380 | 0.0034–0.5000 | 0.1978 | 0.0034–0.3750 | 682 |
| 3A | 481 | 0.0039 | 0.2015 | 0.0034–0.5000 | 0.2768 | 0.0068–0.5000 | 0.2257 | 0.0068–0.3750 | 396 |
| 3B | 744 | 0.0038 | 0.1494 | 0.0034–0.5000 | 0.2214 | 0.0068–0.5000 | 0.1852 | 0.0068–0.3750 | 545 |
| 4A | 387 | 0.0040 | 0.1902 | 0.0034–0.4914 | 0.2708 | 0.0068–0.4999 | 0.2222 | 0.0068–0.3749 | 317 |
| 4B | 403 | 0.0037 | 0.1924 | 0.0034–0.4931 | 0.2737 | 0.0068–0.4999 | 0.2234 | 0.0068–0.3750 | 317 |
| 5A | 615 | 0.0038 | 0.1869 | 0.0034–0.4966 | 0.2663 | 0.0068–0.5000 | 0.2185 | 0.0068–0.3750 | 494 |
| 5B | 766 | 0.0037 | 0.2096 | 0.0034–0.5000 | 0.2930 | 0.0068–0.5000 | 0.2389 | 0.0068–0.3750 | 673 |
| 6A | 515 | 0.0036 | 0.2032 | 0.0034–0.4983 | 0.2782 | 0.0068–0.5000 | 0.2269 | 0.0068–0.3750 | 412 |
| 6B | 740 | 0.0040 | 0.1860 | 0.0034–0.4966 | 0.2664 | 0.0068–0.5000 | 0.2191 | 0.0068–0.3750 | 617 |
| 7A | 692 | 0.0043 | 0.1670 | 0.0034–0.4983 | 0.2410 | 0.0068–0.5000 | 0.2001 | 0.0068–0.3750 | 523 |
| 7B | 557 | 0.0040 | 0.1925 | 0.0034–0.4983 | 0.2701 | 0.0068–0.5000 | 0.2216 | 0.0068–0.3750 | 464 |
| Unknown [a] | 1344 | 0.0055 | 0.1784 | 0.0017–0.4983 | 0.2542 | 0.0034–0.5000 | 0.2099 | 0.0034–0.3750 | - |
| A genome | 3853 | 0.0038 | 0.1857 | 0.0017–0.5000 | 0.2623 | 0.0034–0.5000 | 0.2158 | 0.0034–0.3750 | 3095 |
| B genome | 4944 | 0.0038 | 0.1871 | 0.0017–0.5000 | 0.2634 | 0.0034–0.5000 | 0.2162 | 0.0034–0.3750 | 3998 |
| AB whole genome | 8797 | 0.0038 | 0.1865 | 0.0017–0.5000 | 0.2629 | 0.0034–0.5000 | 0.2161 | 0.0034–0.3750 | 7093 |

[a] Unknown, are SNPs not assigned to any of the chromosomes; has no position values in base pair and with no significant match in the genome as well.
[b] Are final set of SNPs defined based on MAF > 5% and used for association and all other downstream analysis.

CI (Table 3), we here focused on reporting the GWAS results in relation to CI. A comprehensive result of the GWAS analysis for SEV, RES and CI of each environment and combined data is presented in S4 Table. The phenotypic variation $R^2$ explained by all six SNPs on chromosome 1B ranged from 51.9–53.9% for CHD-16, 52.1–58.5% for KUL-16, 51.1–52.1% for MER-15, 54.8–56.9% for MER-16 and 62.6–64.0% for BLUE-all (Table 6). MAF ranged from 14.2% to 31.7% with the highest for AX-95171339and the lowest for AX-94856684. The lowest (2.13E-08) significant FDR_Adjusted_P-value was exhibited by AX-94730403 on chromosome 1B. The resistance allele frequency (RAF) ranged from 14.0% to 71.0% among the panel with

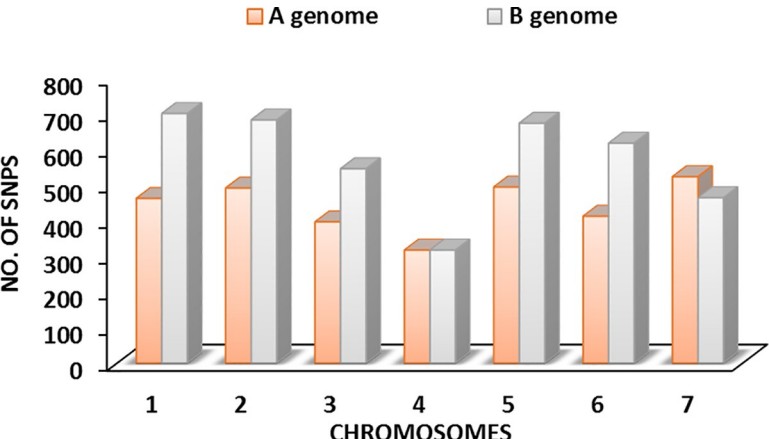

**Fig 4. Chromosomal distribution of 7093 SNPs across genome A and B.** They are a subset of the 8797 SNPs (Table 4) and are selected based on their MAF values of > 5% for all subsequent analysis.

marker AX-94856684 (on chromosome 2B) attaining the lowest value while AX-95238778 and AX-95096041(on chromosome 1B) having the highest value.

Variation of phenotypic values of resistant and susceptible alleles was also assessed with Two-Sample T-test (assuming equal variances). This resulted in a highly significant difference between the resistant and susceptible allele carrying individuals (Table 7).

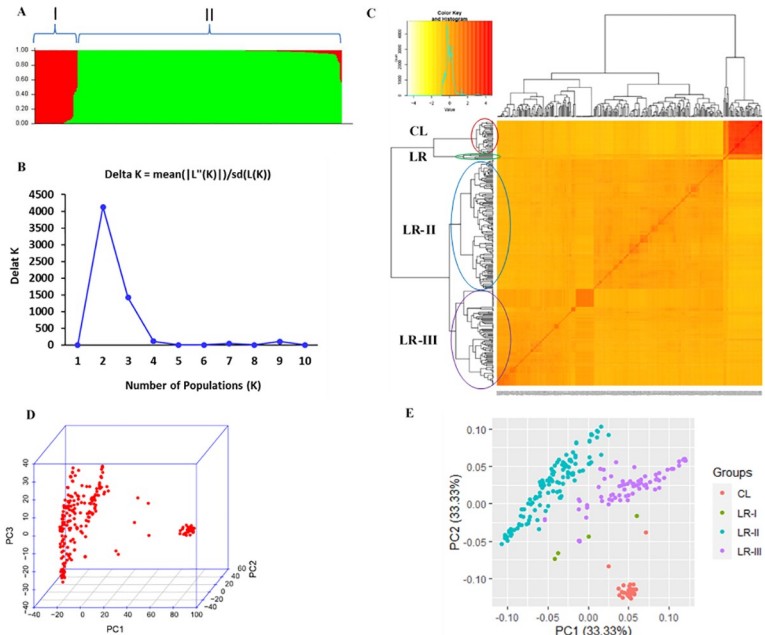

**Fig 5. Population structure and relatedness among the 293 accessions used for the GWAS.** A) population structure plot as reveled by STRUCTURE analysis. B) Delta K plot from STRUCTRUE HARVESTER analysis. C) 293x293 Kinship matrix plot using genetic similarity matching where outside the matrix is a clustering tree of the panel in to 2 main groups (in global view) while a bit of detailed view gave four clusters: CL = Cultivars and three Land Races sub-groups (LR-I, LR-II and LR-III). At the top left corner of the plot is the distribution of estimated kinship values. D) 3D PCA plot of the first three principal components where improved varieties still stood out in a distinct group at the very right end in the plot while the rest grouped in a similar way to the Kinship plot. E) 2D PCA plot elaborating sub-groups along with the variation explained by the components.

**Table 5. Linkage disequilibrium estimate among Ethiopian durum wheat panel.**

| Classes (Mbp) | Number of pairs | No of Significant* Paris | % of Significant Paris | Mean r^2 | Mean of r^2>0.2 |
|---|---|---|---|---|---|
| 0–10 | 9996 | 1367 | 13.68% | 0.28 | 0.36 |
| 10–20 | 9979 | 795 | 7.97% | 0.16 | 0.29 |
| 20–30 | 9968 | 930 | 9.33% | 0.14 | 0.29 |
| 30–40 | 9956 | 996 | 10.00% | 0.13 | 0.29 |
| 40–50 | 9947 | 1146 | 11.52% | 0.13 | 0.29 |
| 50–60 | 9896 | 1317 | 13.31% | 0.13 | 0.29 |
| 60–70 | 9880 | 1355 | 13.71% | 0.13 | 0.30 |
| >70 | 558264 | 168665 | 30.21% | 0.11 | 0.34 |
| Total | 627886 | 176571 | 28.12% | 0.11 | 0.34 |

*significance in LD was declared at $P \leq 0.001$ where $P$ represents probability of Disequilibrium (pDiseq).

## Other genes in the vicinity of SNPs associated with the Pst resistance

Further investigation on the genomic region encompassing the significantly associated SNPs in all the four chromosomes revealed 44 genies of various descriptive functions (Table 8). Thirty-six of the genes were identified In approximately 10.5Mbp (325818004–336274839) genomic region encompassing the identified resistance associated SNPs on chromosome 1B. Four of these genes (MATH domain containing protein (TraesCS1B01G180400), Alpha-galactosidase (TraesCS1B01G181700), Chloroplast inner envelope protein putative, expressed (TraesCS1B01G182300) and Plant basic secretory family protein (TraesCS1B01G182700)) were redundantly found. SNP AX-95171339, which is found close to the outer most of this region is very closely associated to Pentatricopeptide repeat-containing protein (TraesCS1B01G179700) while SNP AX-94436448 is flanked by DNA-directed RNA polymerase subunit (TraesCS1B01G179900) and Peptide chain release factor 2 (TraesCS1B01G180000). The ABC transporter gene family (TraesCS1B01G181200) which is known to confer durable

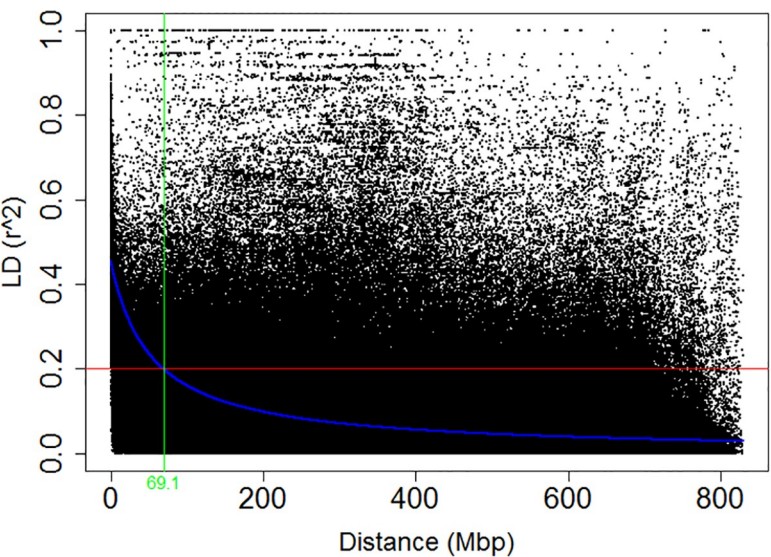

**Fig 6. Genome-wide linkage disequilibrium as dictated by physical distance.** Average pair-wise inter-SNP LD ($r^2$) values plotted against physical distance in base pairs based on the wheat reference genome RefSeq v.1.0. The red line indicates the threshold LD.

**Table 6. SNPs significantly associated with Yellow rust resistance identified by GWAS analysis.**

| SNP ID | Chr | Position[a] | SNP[b] | RAF | P.value | MAF | $R^{2}$ [c] | FDR_ Adjusted _P-values | Effect |
|---|---|---|---|---|---|---|---|---|---|
| **CI_BLUE_CHD-16** | | | | | | | | | |
| AX-94856684 | 2B | 88928792 | T/**C** | 0.14 | 1.67E-05 | 0.142 | 0.522 | 2.95E-02 | -0.593 |
| AX-95171339 | 1B | 325818519 | **T**/C | 0.68 | 7.15E-06 | 0.317 | 0.525 | 1.69E-02 | -0.168 |
| AX-94436448 | 1B | 326782345 | **T**/C | 0.70 | 3.60E-05 | 0.300 | 0.519 | 3.69E-02 | -0.152 |
| AX-95238778 | 1B | 327358600 | **T**/G | 0.71 | 4.17E-05 | 0.294 | 0.519 | 3.69E-02 | -0.151 |
| AX-95096041 | 1B | 327577683 | T/**C** | 0.71 | 4.17E-05 | 0.294 | 0.519 | 3.69E-02 | 0.151 |
| AX-94730403 | 1B | 328938869 | T/**C** | 0.20 | 1.55E-07 | 0.201 | 0.539 | 1.10E-03 | 0.227 |
| AX-94427201 | 1B | 328942601 | **C**/G | 0.21 | 1.63E-06 | 0.203 | 0.530 | 5.77E-03 | 0.190 |
| AX-94482796 | 1A | 352861762 | **T**/C | 0.20 | 2.54E-05 | 0.195 | 0.520 | 3.61E-02 | 0.155 |
| **CI_BLUE_KUL-16** | | | | | | | | | |
| AX-94438404 | 5A | 9840211 | T/**C** | 0.15 | 5.88E-05 | 0.152 | 0.521 | 4.63E-02 | 0.332 |
| AX-94460229 | 5A | 9840244 | T/**C** | 0.15 | 5.88E-05 | 0.152 | 0.521 | 4.63E-02 | 0.332 |
| AX-95171339 | 1B | 325818519 | **T**/C | 0.68 | 2.17E-11 | 0.317 | 0.577 | 7.70E-08 | -0.332 |
| AX-94436448 | 1B | 326782345 | **T**/C | 0.70 | 3.18E-09 | 0.300 | 0.557 | 5.63E-06 | -0.288 |
| AX-95238778 | 1B | 327358600 | **T**/G | 0.71 | 8.45E-09 | 0.294 | 0.554 | 9.99E-06 | -0.281 |
| AX-95096041 | 1B | 327577683 | T/**C** | 0.71 | 8.45E-09 | 0.294 | 0.554 | 9.99E-06 | 0.281 |
| AX-94730403 | 1B | 328938869 | T/**C** | 0.20 | 3.00E-12 | 0.201 | 0.585 | 2.13E-08 | 0.402 |
| AX-94427201 | 1B | 328942601 | **C**/G | 0.21 | 3.91E-11 | 0.203 | 0.575 | 9.23E-08 | 0.361 |
| AX-94648330 | 1B | 336210294 | T/**C** | 0.16 | 2.21E-05 | 0.160 | 0.524 | 2.24E-02 | 0.295 |
| **CI_BLUE_MER-15** | | | | | | | | | |
| AX-95171339 | 1B | 325818519 | **T**/C | 0.68 | 1.16E-05 | 0.317 | 0.516 | 2.75E-02 | -0.212 |
| AX-94436448 | 1B | 326782345 | **T**/C | 0.70 | 4.28E-05 | 0.300 | 0.511 | 5.05E-02 | -0.196 |
| AX-95238778 | 1B | 327358600 | **T**/G | 0.71 | 2.61E-05 | 0.294 | 0.513 | 3.71E-02 | -0.203 |
| AX-95096041 | 1B | 327577683 | T/**C** | 0.71 | 2.61E-05 | 0.294 | 0.513 | 3.71E-02 | 0.203 |
| AX-94730403 | 1B | 328938869 | T/**C** | 0.20 | 5.86E-06 | 0.201 | 0.519 | 2.08E-02 | 0.261 |
| AX-94427201 | 1B | 328942601 | **C**/G | 0.21 | 3.56E-06 | 0.203 | 0.521 | 2.08E-02 | 0.262 |
| **CI_BLUE_MER-16** | | | | | | | | | |
| AX-95171339 | 1B | 325818519 | **T**/C | 0.68 | 1.02E-11 | 0.203 | 0.569 | 5.40E-08 | 0.458 |
| AX-94436448 | 1B | 326782345 | **T**/C | 0.70 | 1.52E-11 | 0.201 | 0.568 | 5.40E-08 | 0.466 |
| AX-95238778 | 1B | 327358600 | **T**/G | 0.71 | 3.95E-11 | 0.317 | 0.564 | 9.34E-08 | -0.387 |
| AX-95096041 | 1B | 327577683 | T/**C** | 0.71 | 1.49E-09 | 0.294 | 0.549 | 2.12E-06 | 0.350 |
| AX-94730403 | 1B | 328938869 | T/**C** | 0.20 | 1.49E-09 | 0.294 | 0.549 | 2.12E-06 | -0.350 |
| AX-94427201 | 1B | 328942601 | **C**/G | 0.21 | 2.09E-09 | 0.300 | 0.548 | 2.47E-06 | -0.345 |
| **CI_BLUE_All** | | | | | | | | | |
| AX-95171339 | 1B | 325818519 | **T**/C | 0.68 | 1.62E-10 | 0.317 | 0.637 | 3.82E-07 | -0.282 |
| AX-94436448 | 1B | 326782345 | **T**/C | 0.70 | 5.90E-09 | 0.300 | 0.626 | 9.25E-06 | -0.252 |
| AX-95238778 | 1B | 327358600 | **T**/G | 0.71 | 7.83E-09 | 0.294 | 0.625 | 9.25E-06 | -0.251 |
| AX-95096041 | 1B | 327577683 | T/**C** | 0.71 | 7.83E-09 | 0.294 | 0.625 | 9.25E-06 | 0.251 |
| AX-94730403 | 1B | 328938869 | T/**C** | 0.20 | 5.04E-11 | 0.201 | 0.641 | 2.61E-07 | 0.335 |
| AX-94427201 | 1B | 328942601 | **C**/G | 0.21 | 7.36E-11 | 0.203 | 0.640 | 2.61E-07 | 0.311 |

[a] position in base pair of each SNP based on wheat genome reference sequence (Refseq.v1).

[b] SNP nucleotides in bold font are the resistant alleles while the other is the alternate susceptible allele.

[c] $R^2$: Coefficient of Determination; that is the proportion of phenotypic effect explained by the model for each significant locus.

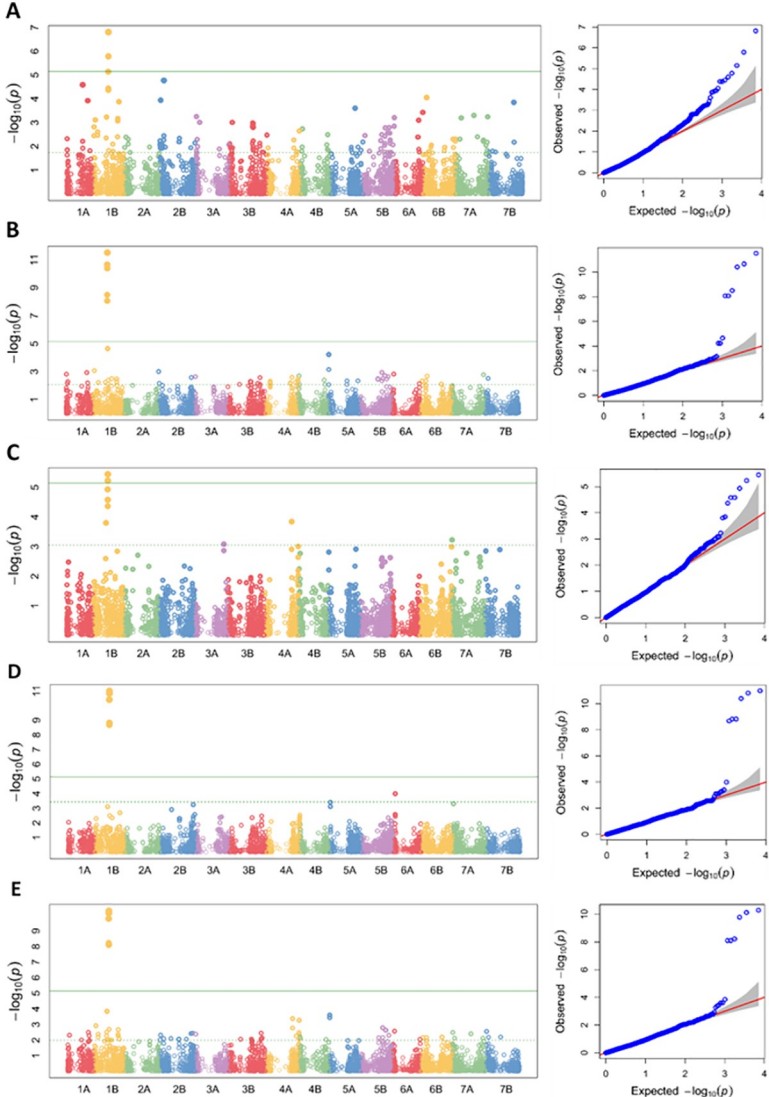

**Fig 7. Genome-wise Manhattan and QQ plots of GWAS of yellow rust resistance in Ethiopian durum wheat.**
Results represent association analysis of CI data from 293 durum wheat accessions at CHD-16 (A), KUL-16 (B) MER-15 (C), MER-16 (D) and combined over the four environments (E). The panel were genotyped with Breeders' 35K Axiom Array for wheat.

resistance to multiple fungal pathogens in wheat [69] is also very closely located with AX-95096041. Similarly, the disease resistance protein RPM1 (TraesCS1B01G182900) belonging to known disease resistance protein families (NBS-LRR class) was also found proximal to SNP AX-94427201 besides the Receptor kinase. The Other 8 genes were identified as flanks in the vicinity of the remaining 4 SNPs on chromosome 1A, 5A and 2B with the Protein kinase flanking the SNP AX-94856684 approximately 2.2 Kb away on chromosome 2B (Table 8).

## Discussion

### Phenotypic variability in resistance to Pst

The average field response to *Pst* of accessions ranged from very low to high among the environments. This is very clear when examining particularly the result of CHD-15 and KUL-15

**Table 7. Single locus-based variation of phenotypic values of resistant and susceptible alleles as resulted from two-sample T-test assuming equal variances.**

| SNP ID | Chr. | SNP Alleles [a] | Mean CI | Observations | df | t Stat | P(T< = t) two-tail[b] |
|---|---|---|---|---|---|---|---|
| AX-94482796 | 1A | **T** / C | 9.96/7.66 | 57/236 | 291 | 1.08 | 2.82E-01* |
| AX-95171339 | 1B | **T** / C | 4.26/16.36 | 200/93 | 291 | -7.20 | 5.06E-12*** |
| AX-94436448 | 1B | **T** / C | 4.70/16.04 | 205/88 | 291 | -6.57 | 2.38E-10*** |
| AX-95238778 | 1B | **T** / G | 4.66/16.41 | 207/86 | 291 | -6.78 | 6.51E-11*** |
| AX-95096041 | 1B | **C** / T | 4.66/16.41 | 207/86 | 291 | -6.78 | 6.51E-11*** |
| AX-94730403 | 1B | **C** / T | 3.40/9.32 | 60/233 | 291 | -2.85 | 4.70E-03*** |
| AX-94427201 | 1B | **C** / G | 3.21/9.36 | 60/233 | 291 | -2.97 | 3.25E-03*** |
| AX-94648330 | 1B | **C** / T | 4.73/8.75 | 47/246 | 291 | -1.75 | 8.16E-02ns |
| AX-94856684 | 2B | **T** / C | 8.24/7.30 | 252/41 | 291 | 0.38 | 7.01E-01ns |
| AX-94438404 | 5A | **C** / T | 2.39/9.11 | 44/249 | 291 | -2.87 | 4.41E-03*** |
| AX-94460229 | 5A | **C** / T | 2.39/9.11 | 44/249 | 291 | -2.87 | 4.41E-03*** |

[a] SNP nucleotide in bold font & underlined are the resistant alleles while the other is the alternate susceptible allele; the values of Mean CI and Observations before and after the "/" sign stands for the corresponding SNP Alleles.
[b] all P values marked with '***'signs shows the difference of phenotypic value in terms of CI is clearly significant between the resistant and susceptible allele carrying lines; for those with p values of '**ns**' no difference between the resistance and susceptible alleles carrying lines.

environments for which the mean SEV, RES and CI values of susceptible genotypes are 1.1, 0.3, 1.1 and 2.6, 0.3, 2.5 respectively (Table 1; Fig 1A–1C). Notably, field response at Meraro was higher because the location is known to be one of the hot spot sites for *Pst* infestation, and hence usually used as a stripe rust test site. However, considering the reaction of the susceptible genotypes, the average SEV (88.5%), RES (1.0), CI (87.5) in 2015 was still lower than it was in 2016 where SEV = 98.0%, RES = 1.0 and CI = 98.0. This might be attributed to a shortage of moisture in 2015. It is known that establishment of *Pst* infection in the field is highly dependent on available moisture and cool night temperatures which ultimately affects disease development. Ninety-one (31.1%) of the panel consistently appeared as resistant across all the four environments (S3 Table) which is explained by the broad-spectrum resistance of the accessions for the *Pst* race composition at the environments. Overall, the reaction data was unevenly distributed for the SEV, RES & CI at CHD-15 and KUL-15 environments which were excluded from all downstream analyses to avoid misleading generalization while it was relatively okay for the CHD-16, KUL-16, MER-15 & MER-16. Variance component due to blocking nested within replication was not significant indicating variation due to nesting or incomplete blocking was negligible. On the other hand, genotype by environment interactions have significantly varied at combined level which obviously is due to variation in disease pressure among the environments.

Disease reaction data (SEV, RES & CI) have shown relatively considerable correlations within location between years for Meraro although not that high to be significant. This is in agreement with the lower disease pressure in 2o15 as compared to 2016. Overall correlation of SEV, RES and CI among the four environments is also low which is another confirmation for the varying level of disease prevalence among the environments with a little bit of similarity for Merao test location (Table 3; Fig 1A–1C). On the other hand, correlations for Inter-disease reaction data combinations (SEV vs RES; SEV vs CI; RES vs CI) among environments were very high, ranging from 0.91–0.96 for SEV vs RES; 0.96–0.98 for SEV vs CI and 0.91–0.95 for RES vs CI. This provided the basis for performing the GWAS analysis on any of the three disease reaction data although CI is preferred as it is a combined representation of SEV and RES.

**Table 8. Genes and gene models[a] reported with in the identified Yr resistance associated regions and close to the significant SNP.**

| SNP ID | SNP position | Chromosome | Gene/gene model position | | | Gene names | Description of genes |
|---|---|---|---|---|---|---|---|
| | | | Start | End | strand | | |
| | | chr1A | 352853002 | 352853059 | + | TraesCS1A01G195300 | Phenylalanine—tRNA ligase beta subunit |
| AX-94482796 | 352861762 | chr1A | | | | | |
| | | chr1A | 353253624 | 353254735 | + | TraesCS1A01G195400 | Poly [ADP-ribose] polymerase |
| | | chr1B | 325818004 | 325818017 | - | TraesCS1B01G179600 | Protease inhibitor/seed storage/lipid transfer family protein |
| AX-95171339 | 325818519 | chr1B | | | | | |
| | | chr1B | 325977067 | 325978058 | + | TraesCS1B01G179700 | Pentatricopeptide repeat-containing protein |
| | | chr1B | 326003456 | 326003632 | - | TraesCS1B01G179800 | GRAM domain-containing protein / ABA-responsive protein-related |
| | | chr1B | 326763409 | 326763808 | + | TraesCS1B01G179900 | DNA-directed RNA polymerase subunit beta |
| AX-94436448 | 326782345 | chr1B | | | | | |
| | | chr1B | 326783032 | 326783094 | + | TraesCS1B01G180000 | Peptide chain release factor 2 |
| | | chr1B | 326912448 | 326912523 | + | TraesCS1B01G180100 | Ubiquitin carboxyl-terminal hydrolase 12 |
| | | chr1B | 326925257 | 326927809 | - | TraesCS1B01G180200 | Subtilisin-like protease |
| | | chr1B | 327023284 | 327023426 | - | TraesCS1B01G180300 | Protein FLX-like 3 |
| | | chr1B | 327227211 | 327227232 | + | TraesCS1B01G180400 | MATH domain containing protein |
| | | chr1B | 327338934 | 327338955 | + | TraesCS1B01G180500 | MATH domain containing protein |
| | | chr1B | 327347090 | 327347111 | + | TraesCS1B01G180600 | Ubiquitin carboxyl-terminal hydrolase-like protein |
| | | chr1B | 327354240 | 327354261 | + | TraesCS1B01G180700 | Nucleolar complex protein 2 |
| AX-95238778 | 327358600 | chr1B | | | | | |
| | | chr1B | 327360657 | 327361778 | + | TraesCS1B01G180800 | pH-response regulator protein palA/RIM20 |
| | | chr1B | 327478353 | 327479021 | + | TraesCS1B01G180900 | ALG-2 interacting protein X |
| | | chr1B | 327575252 | 327575440 | - | TraesCS1B01G181000 | KH domain-containing protein |
| AX-95096041 | 327577683 | chr1B | | | | | |
| | | chr1B | 327831608 | 327831635 | + | TraesCS1B01G181100 | Katanin p80 WD40 repeat-containing subunit B1 homolog |
| | | chr1B | 327842474 | 327843500 | - | TraesCS1B01G181200 | E3 ubiquitin-protein ligase Hakai |
| | | chr1B | 327958104 | 327958448 | - | TraesCS1B01G181300 | **ABC transporter G family member** |
| | | chr1B | 328104651 | 328105660 | + | TraesCS1B01G181400 | DNA topoisomerase family |
| | | chr1B | 328436238 | 328436336 | + | TraesCS1B01G181500 | Plant/F9H3-4 protein |
| | | chr1B | 328642492 | 328642600 | - | TraesCS1B01G181600 | WAT1-related protein |
| | | chr1B | 328649327 | 328649672 | + | TraesCS1B01G181700 | Hsp20/alpha crystallin family protein |
| | | chr1B | 328820817 | 328820900 | - | TraesCS1B01G181800 | Alpha-galactosidase |
| | | chr1B | 328827925 | 328828332 | + | TraesCS1B01G181900 | Serine/threonine protein phosphatase 7 long form isogeny |
| | | chr1B | 328938856 | 328938960 | - | TraesCS1B01G182000 | Alpha-galactosidase |
| AX-94730403 | 328938869 | chr1B | | | | | |
| AX-94427201 | 328942601 | chr1B | | | | | |
| | | chr1B | 328988642 | 328988845 | + | TraesCS1B01G182100 | Ser/Thr protein phosphatase family protein, expressed |
| | | chr1B | 329202234 | 329202623 | - | TraesCS1B01G182200 | Aquaporin |
| | | chr1B | 329295598 | 329296214 | - | TraesCS1B01G182300 | **Receptor kinase** |
| | | chr1B | 329372867 | 329373330 | + | TraesCS1B01G182400 | Chloroplast inner envelope protein, putative, expressed |
| | | chr1B | 329478217 | 329478761 | + | TraesCS1B01G182500 | Chloroplast inner envelope protein, putative, expressed |
| | | chr1B | 329488988 | 329489122 | - | TraesCS1B01G182600 | G-patch domain containing protein, expressed |

(*Continued*)

**Table 8.** (Continued)

| SNP ID | SNP position | Chromosome | Gene/gene model position | | | Gene names | Description of genes |
|---|---|---|---|---|---|---|---|
| | | | Start | End | strand | | |
| | | chr1B | 329490934 | 329491422 | + | TraesCS1B01G182700 | Ubiquitin |
| | | chr1B | 329713781 | 329714488 | + | TraesCS1B01G182800 | Plant basic secretory family protein |
| | | chr1B | 329763404 | 329763604 | + | TraesCS1B01G182900 | Peroxidase |
| | | **chr1B** | **329941622** | **329942682** | **+** | TraesCS1B01G183000 | **Disease resistance protein RPM1[b]** |
| | | chr1B | 329960233 | 329960910 | - | TraesCS1B01G183100 | Plant basic secretory family protein |
| | | chr1B | 336207098 | 336207277 | + | TraesCS1B01G188000 | Monothiol glutaredoxin |
| AX-94648330 | 336210294 | chr1B | | | | | |
| | | chr1B | 336273637 | 336274839 | - | TraesCS1B01G188100 | Pentatricopeptide repeat-containing protein |
| | | chr2B | 88928630 | 88928652 | + | TraesCS2B01G121400 | F-box family protein |
| AX-94856684 | 88928792 | chr2B | | | | | |
| | | chr2B | 88930677 | 88930994 | - | TraesCS2B01G121500 | **Protein kinase** |
| | | chr5A | 9834951 | 9835031 | + | TraesCS5A01G014500 | Enolase |
| AX-94460229 | 9840244 | chr5A | | | | | |
| AX-94438404 | 9840211 | chr5A | | | | | |
| | | chr5A | 9840483 | 9840678 | - | TraesCS5A01G014600 | WD-repeat protein, putative |

[a] The Genes/gene model list is extracted from the wheat gnome annotation file (IWGSCv1.0_UTR.HC.canonicalcds) based on the recently availed wheat genome reference sequence Refseqv1.

[b] Disease resistance protein (RPM1) is one of the widely reported genes known to have direct involvement in plant defense system against pathogens. It is about 1 Mbp away from the nearest significant SNP (i.e AX-94427201) and 6.27 Mbp from another closely located significant SNP (AX-94648330).

## Population structure, relatedness and LD

The population structure analysis plot clustered the panel into two distinct groups where members of group I are mainly (36/39) improved durum cultivars while that of group II contains all the landraces besides three cultivars. This is in agreement with the structure harvester output that suggests K = 2 is the most likely grouping value of the panel. The presence of the two groups in the population structure and further subgrouping in the Kinship has not shown any significant correlation with the pattern of the phenotypic values (resistance reaction) among the panel. Consequently, the data led to identification of a true and acceptable marker-trait association for the resistance as opposed to the discovery of false positive association.

## Analysis of the resistance loci identified on chromosomes

It looks that the significant genotypic variance among the panel was reflected in the identification of significant marker-trait associations at various level. Similar sets of SNP association were identified both at single and combined environment data analyses with few exceptions suggesting existence of main wide spectrum vs minor environment specific effectiveness of the identified resistance loci. Obviously, the consistence occurrence of the six significantly associated SNPs at all levels of analyses probably has to do with a wide spectrum effectiveness of the resistance gene/genes underlying the loci as well. On the other hand, the five significant SNPs identified as location specific (2 at CHD and 3 at KUL) associations could mean the presence of site specific *Pst* race accompanied by race specific genes relative to the other sites.

Our data suggests that chromosome 1B is an important contributor of loci significantly associated with the resistance as seven of the eleven identified SNPs are located on it. Particularly in all single environment and BLUE_all GWAS, all six associated SNPs were from this chromosome. Several genes associated with yellow rust resistance in wheat have been reported on chromosome 1B from multiple GWAS studies. This highlights its usefulness and why efforts to further define these loci are warranted. So far, 83 *Pst* resistance genes have been designated [31] and many QTLs identified in wheat have been reported on chromosome 1B [70]. *Yr10* [71], *Yr9* [72], *YrAlp* [31], *Yr15* [73], *YrH52* [74], *Yr64*, *Yr65* and *Yr24/Yr26* [75], *YrExp1* [31], *Yr29/Lr46* [76] are some of the known YR genes identified on chromosome 1B and derived from wild relatives and cultivars. Interestingly, the recently cloned *Yr15* gene is in position 547Mb in CS on chromosome 1B and the top markers identified in the current study are in position 325–336 Mb suggesting that the identified loci are different from *Yr15*. One additional SNP was identified on chromosome 1A while the other two came from chromosome 5A and another one from chromosome 2B. Several yellow rust QTLs and *Yr* genes are mapped on these chromosomes including *Yr5* on chromosome 2B [77] which could be amongst the few that are effective against Ethiopian *Pst* races [44, 78].

The genes that are found in the proximity of the significant loci (ABC Transporter gene family, RPM1, Receptor kinases and Protein kinases) shades light on the functional association of the identified regions in plant defences against pathogens. The ABC transporter gene family is among the known ones involved in the plant defense system conferring durable resistance to multiple fungal pathogens in wheat [69]. They usually act as a transporter for different molecules across biological membranes and are involved in a diverse range of biological processes [79]. RPM1 is an NBS-LRR protein from *Arabidopsis thaliana* that confers resistance to *Pseudomonas syringae* expressing either avrRpm1 or avrB [80]. It is also characterized as a peripheral membrane protein that likely resides on the cytoplasmic face of the plasma membrane [80]. Receptor kinases also called Receptor-like kinases (RLks) are among the pattern recognitions receptors that play an important role in plant immunity apart from growth and development [81]. During plant-pathogen interaction, they function as a part of a multiprotein complexes at the cell surface which detect microb-and host-derived molecular patterns as the first layer of inducible defence [82]. Likewise, Protein kinases plays a pivotal role in the activation of plant defence mechanisms through signalling during pathogen recognition at the start of the host-pathogen interaction [83]. In general, the fact that these defence related genes are located in the vicinity of the identified loci suggests future attentions towards functional studies.

To the best of our information, this is the first GWAS study in Ethiopia on Ethiopian durum wheat across the three test locations for identification of marker trait association (MTAs) for *Pst* resistance. However, a similar study conducted on Ethiopian durum wheat in the USA led to the identification of 12 loci associated with resistance to *Pst* on seven chromosomes of which chromosome 1B is one of them besides chromosomes 1A, 2BS, 3BL, 4AL, 4B and 5AL [17]. On the other hand, Zegeye [84] carried out a GWAS on synthetic hexaploid wheat at Meraro and Arsi Robe and reported a total of 38 SNPs on 18 genomic regions associated with adult plant resistance. Some of these reported genomic regions are also identified on chromosome 1B besides 1A, 2B and 5A which is in agreement with the current study. So, similarities of this genomic regions in response to *Pst* resistance across various similar studies signifies the potential usefulness of the genomic regions for wheat resistance improvement.

## Conclusion

This study identified 11 SNPs which significantly associated with resistance to *Pst* at adult plant stage and defined at least five genomic Loci (1 on 1A, 2 on 1B, 1 on 2B and 1 on 5A)

across the study panel. Six of the SNPs (AX-95171339, AX-94436448, AX-95238778, AX-95096041, AX-94730403 & AX-94427201) defined the first regions on chromosome 1B. These SNPs, as they were consistently identified at each environment and combined data the region can be considered as the major genomic locus which might contain combinations of genes conferring resistance to *Pst* across all tested environments. On the other hand, SNP AX-94648330 on 1B, a haplotype of two closely linked SNPs (AX-94438404 & AX-94460229) on 5A, AX-94482796 on 1A and AX-94856684 on 2B identified only at KUL or CHD defined four additions loci. The fact that these SNPs are identified either only at CHD or KUL but not at MER may indicate the presence of resistance gene/genes effective to location specific *Pst* races. This calls for a separate consideration in future breeding strategies for durable *Pst* resistance enhancement in wheat which should take into account race-specificity. Disease resistance related and other genes such ABC transporter, RPM1 and the Receptor kinases and Protein kinases have been found in the vicinity of these resistance associated loci. These genes can be targeted in any functional study aiming to reveal the actual genes underlying the associated loci. The study also identified effective sources of resistance to Ethiopian *Pst* races in Ethiopian durum wheat landraces that can be used, alongside the markers identified here, to transfer the loci into adapted cultivars to provide resistance against *Pst*. However, the diagnostic value of the identified SNPs needs to be further investigated and validated in an independent germplasm including phenotypic response test to the pathogen. In general, the identified SNPs/resistance loci, coupled with the identification of multi-environment stable genotypes for resistance, will enhances the fight towards mitigation of *Pst* as it presents a double layer challenge both to the wide spectrum and site specific virulent *Pst* races.

## Supporting information

**S1 Table. List of catalogued *Yr* genes derived from various sources.**
(CSV)

**S2 Table. Ethiopian durum wheat landraces and cultivars used for the GWAS.**
(CSV)

**S3 Table. Genotypes consistently resistant across the four environments.**
(CSV)

**S4 Table. Association analysis result for all single and combined environment data.**
(CSV)

## Acknowledgments

The authors are grateful to Kulumsa (KARC) and Debre Zeit Agricultural Research Centre (DZARC) for kindly provision of experimental plot & cultivars; Ethiopian Biodiversity Institute (EBI) and EOSA (Ethio-Organic Seed Action) for their kind provision of the landrace accessions. Our gratitude also goes to wheat breeders, pathologists and technical assistances at Kulumsa and Debre Zeit research centers for their assistance in field work. We are so grateful to all members of the Uauy group; Brande Wulf, Sanu Arora, Clare Lewis and Luzie Wingen at the JIC for their technical support and encouraging guidance during laboratory work.

## Author Contributions

**Conceptualization:** Sisay Kidane Alemu.

**Data curation:** Sisay Kidane Alemu.

**Formal analysis:** Sisay Kidane Alemu.

**Funding acquisition:** Kassahun Tesfaye, Cristobal Uauy.

**Investigation:** Sisay Kidane Alemu.

**Methodology:** Sisay Kidane Alemu, Ayele Badebo Huluka, Cristobal Uauy.

**Project administration:** Cristobal Uauy.

**Resources:** Cristobal Uauy.

**Supervision:** Ayele Badebo Huluka, Kassahun Tesfaye, Teklehaimanot Haileselassie, Cristobal Uauy.

**Visualization:** Sisay Kidane Alemu.

**Writing – original draft:** Sisay Kidane Alemu.

**Writing – review & editing:** Sisay Kidane Alemu, Ayele Badebo Huluka, Kassahun Tesfaye, Teklehaimanot Haileselassie, Cristobal Uauy.

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
