## [Decision Letter · Decision Letter 0]

24 Dec 2020

PONE-D-20-37036

Genome-wide association mapping identifies Yellow Rust resistance locus in Ethiopian Durum Wheat germplasm

PLOS ONE

Dear Dr. Alemu,

Thank you for submitting your manuscript to PLOS ONE. After careful consideration, we feel that it has merit but does not fully meet PLOS ONE’s publication criteria as it currently stands. Therefore, we invite you to submit a revised version of the manuscript that addresses the points raised during the review process.

We look forward to receiving your revised manuscript.

Kind regards,

Aimin Zhang, Ph.D.

Academic Editor

PLOS ONE

Journal Requirements:

Reviewers' comments:

Reviewer's Responses to Questions

**Comments to the Author**

1. Is the manuscript technically sound, and do the data support the conclusions?

Reviewer #1: Partly

Reviewer #2: Partly

2. Has the statistical analysis been performed appropriately and rigorously? 

Reviewer #1: N/A

Reviewer #2: Yes

3. Have the authors made all data underlying the findings in their manuscript fully available?

Reviewer #1: No

Reviewer #2: Yes

4. Is the manuscript presented in an intelligible fashion and written in standard English?

Reviewer #1: Yes

Reviewer #2: Yes

5. Review Comments to the Author

Reviewer #1: This work conducted a genome wide association study of yellow rust resistance on 300 Ethiopian durum wheat accessions. The results are helpful to resistance breeding. My suggestions:

(1) please show the phenotype data of spreader to judge the quality of field resistance evaluation;

(2) please adding the phenotype data of analyzed wheat lines in Table S1, which is imporatant information for further genetics and breeding stduies.

(2) For SEV, 100% of the accessions were classified as resistant (0≤SEV≤10) at CHD_15 and 99.7% at KUL_15 . For RES, 96.6% of the accessions fell under resistant class at CHD_15 and 97.63% at KUL_15. The low Pst infestation can not provide useful information in following analysis, but makes the analysis complex and confusion. I suggested to discard the data of the two experiments in further analysis.

Other a few corrections in the attached file.

Reviewer #2: This manuscript reported genome-wide association mapping of yellow rust resistance locus in large number of Ethiopian Durum Wheat germplasms. The author identified 12 SNPs significantly associated with yellow rust resistance across four chromosomes, meanwhile this provides SNPs for tracking the QTL associated with yellow rust resistance in durum wheat improvement programs. However, several revision should be done before the manuscript can be accepted for publication.

1.After the introduction of Yr genes identification, please add some description of Yr genes that have been identified in Durum wheat in detail.

2.Please added some introduction of MAS of the Yr genes in Introduction.

3.Introduction part is too long. Please delete some unessential description.

4.If the germplasm resources in this study can be provided to the public, and where we can get?

5.In line 269, the the number “829100000” should be changed in to “829,100,000”

6.The Tables in this study should be changed into the “Three-line table” format.

7.If the elite resistant loci identified in this study have been transferred into susceptible cultivars in the production? If yes, please state it in Discussion

8.The References have Non-uniform format, please revised them meticulously.

6. PLOS authors have the option to publish the peer review history of their article (what does this mean?). If published, this will include your full peer review and any attached files.

Reviewer #1: No

Reviewer #2: **Yes: **Pengtao Ma

---

## [Author Response · Author response to Decision Letter 0]

6 Apr 2021

I have uploaded the response to each points raised by the academic editor and the reviewers in this revised submission as directed by the Academic Editor. (file name :"Response to Reviewers")

---

## [Decision Letter · Decision Letter 1]

19 Apr 2021

Genome-wide association mapping identifies Yellow Rust resistance loci in Ethiopian Durum Wheat germplasm

PONE-D-20-37036R1

Dear Dr. Alemu,

We’re pleased to inform you that your manuscript has been judged scientifically suitable for publication and will be formally accepted for publication once it meets all outstanding technical requirements.

Kind regards,

Aimin Zhang, Ph.D.

Academic Editor

PLOS ONE

Additional Editor Comments (optional):

Reviewers' comments:

Reviewer's Responses to Questions

**Comments to the Author**

1. If the authors have adequately addressed your comments raised in a previous round of review and you feel that this manuscript is now acceptable for publication, you may indicate that here to bypass the “Comments to the Author” section, enter your conflict of interest statement in the “Confidential to Editor” section, and submit your "Accept" recommendation.

Reviewer #1: All comments have been addressed

2. Is the manuscript technically sound, and do the data support the conclusions?

Reviewer #1: Yes

3. Has the statistical analysis been performed appropriately and rigorously? 

Reviewer #1: Yes

4. Have the authors made all data underlying the findings in their manuscript fully available?

Reviewer #1: Yes

5. Is the manuscript presented in an intelligible fashion and written in standard English?

Reviewer #1: Yes

6. Review Comments to the Author

Reviewer #1: (No Response)

7. PLOS authors have the option to publish the peer review history of their article (what does this mean?). If published, this will include your full peer review and any attached files.

Reviewer #1: No

---

## [Editor Report · Acceptance letter]

3 May 2021

PONE-D-20-37036R1 

Genome-wide association mapping identifies Yellow Rust resistance loci in Ethiopian Durum Wheat germplasm 

Dear Dr. Alemu:

I'm pleased to inform you that your manuscript has been deemed suitable for publication in PLOS ONE. Congratulations! Your manuscript is now with our production department. 

Kind regards, 

on behalf of

Prof. Aimin Zhang 

Academic Editor

PLOS ONE